Quercivorol as a lure for the polyphagous and Kuroshio shot hole borers, Euwallacea spp. nr. fornicatus (Coleoptera: Scolytinae), vectors of Fusarium dieback

Dodge Christine 1 christine.dodge@email.ucr.edu
Coolidge Jessica 1
Cooperband Miriam 2
Cossé Allard 2
Carrillo Daniel 3
Stouthamer Richard 1
1 Department of Entomology, University of California , Riverside, CA , USA
2 Otis Laboratory, USDA-APHIS , Buzzards Bay, MA , USA
3 Tropical Research and Education Center, University of Florida , Homestead, FL , USA
Huber Dezene
Electronic publication date: 2017 Aug 17
Publication date: 2017
Volume: 5
Electronic Location ID: e3656
Received 2017 Jun 19; Accepted 2017 Jul 14
Copyright: © 2017 Dodge et al.
Copyright year: 2017
Copyright holder: Dodge et al.
License: This is an open access article distributed under the terms of the Creative Commons Attribution License, which permits unrestricted use, distribution, reproduction and adaptation in any medium and for any purpose provided that it is properly attributed. For attribution, the original author(s), title, publication source (PeerJ) and either DOI or URL of the article must be cited.
License URL: https://creativecommons.org/licenses/by/4.0/

Keywords: Fusarium, Euwallacea, Ambrosia beetle, Scolytinae, Quercivorol, Verbenone, Piperitone, Semiochemical, Phytopathogen, Invasive

Funding: California Avocado Commission USDA NIFA Hatch Fund 194617 USDA Farm Bill 14-8130-0238-CA This work was supported in part by grants to RS from the California Avocado Commission, USDA NIFA Hatch Fund 194617, and USDA Farm Bill 14-8130-0238-CA. This research was funded through Section 10007 of the Farm Bill. Mention of a commercial product does not constitute an endorsement or recommendation for its use by the United States Department of Agriculture. There was no additional external funding received for this study. The funders had no role in study design, data collection and analysis, decision to publish, or preparation of the manuscript.

==============================
The polyphagous shot hole borer and Kuroshio shot hole borer, two members of the Euwallacea fornicatus species complex (Coleoptera: Curculionidae: Scolytinae), are invasive ambrosia beetles that harbor distinct species of Fusarium fungal symbionts. Together with the damage caused by gallery construction, these two phytopathogenic Fusarium species are responsible for the emerging tree disease Fusarium dieback, which affects over 50 common tree species in Southern California. Host trees suffer branch dieback as the xylem is blocked by invading beetles and fungi, forcing the costly removal of dead and dying trees in urban areas. The beetles are also threatening natural riparian habitats, and avocado is susceptible to Fusarium dieback as well, resulting in damage to the avocado industries in California and Israel. Currently there are no adequate control mechanisms for shot hole borers. This paper summarizes efforts to find a suitable lure to monitor shot hole borer invasions and dispersal. Field trials were conducted in two counties in Southern California over a span of two years. We find that the chemical quercivorol is highly attractive to these beetles, and perform subsequent field experiments attempting to optimize this lure. We also explore other methods of increasing trap catch and effects of other potential attractants, as well as the deterrents verbenone and piperitone.

Introduction

Fusarium dieback is an emerging plant disease, first reported in Israel in 2009 (Mendel et al., 2012) and in Southern California in 2012 (Eskalen et al., 2012). The disease is caused in part by two plant pathogenic fungi in the genus Fusarium (Ascomycota: Hypocreales), each of which is associated with an ambrosia beetle in the cryptic Euwallacea fornicatus species complex (Coleoptera: Curculionidae: Scolytinae) (Cooperband et al., 2016; Kasson et al., 2013; O’Donnell et al., 2015; Stouthamer et al., 2017). This species complex consists of at least three, and possibly five, morphologically indistinguishable ambrosia beetles from Southeast Asia (Stouthamer et al., 2017). Three members of this complex have invaded the United States: two in Southern California, and one in Florida and Hawaii. Until recently all members were thought to be the tea shot hole borer (TSHB) Euwallacea fornicatus Eichhoff (1868), a serious pest of tea in Sri Lanka (Austin, 1956; Walgama & Pallemulla, 2005). Although morphologically indistinguishable, molecular analyses revealed significant divergence in mitochondrial and nuclear genes of all three beetles (Eskalen et al., 2013; Stouthamer et al., 2017), which were subsequently given different common names to distinguish them. The beetle clade invading Florida and Hawaii is thought to be E. fornicatus sensu stricto, and so is referred to here as the TSHB (Stouthamer et al., 2017). Two distinct invasions occurred in Southern California: the beetles invading the Los Angeles and San Diego areas have been given the common names polyphagous shot hole borer (PSHB; Cooperband et al., 2016; Eskalen et al., 2013), and Kuroshio shot hole borer (KSHB; Stouthamer et al., 2017), respectively. All Fusarium species associated with these shot hole borers (SHB) are members of the ambrosia Fusarium clade in the Fusarium solani species complex, which includes a number of phytopathogens as well as opportunistic pathogens of mammals (Kasson et al., 2013; O’Donnell et al., 2008).

The polyphagous and Kuroshio shot hole borers are ambrosia beetles in the tribe Xyleborini, a large (∼1,300 spp.) tribe consisting solely of haplodiploid inbreeding species (Normark, Jordal & Farrell, 1999). Members of this tribe form obligate mutualisms with specific ambrosia fungi, which they cultivate and feed upon. Ambrosia beetles transport and introduce the fungi to new host trees in the process of boring brood galleries for reproduction. Unlike most ambrosia beetles, which colonize dead or dying trees (Raffa, Phillips & Salom, 1993), PSHB and KSHB attack living, healthy trees, many of which are susceptible to Fusarium dieback. As the Fusarium invades the host tree vascular system, it gradually restricts the flow of water and nutrients (Eskalen et al., 2012). Paired with structural damage caused by beetle gallery formation, this causes branch dieback, from which trees are unable to recover (Eskalen et al., 2012; Mendel et al., 2012).

The PSHB (Euwallacea sp. #1 in O’Donnell et al., 2015) harbors one of the causative agents of Fusarium dieback, Fusarium euwallaceae (Freeman et al., 2013b; AF-2 in O’Donnell et al., 2015). The KSHB (Euwallacea sp. #5 in O’Donnell et al., 2015) vectors the other causative agent, an unnamed Fusarium sp. (AF-12 in O’Donnell et al., 2015). Both SHB also harbor additional fungal symbionts: the PSHB carries Graphium euwallaceae and Paracremonium pembeum (Lynch et al., 2016), and KSHB carries an undescribed Graphium species. It was shown that PSHB can complete their development on Fusarium euwallaceae, but not on other Fusarium species (Freeman et al., 2013a). Additionally, PSHB can complete their development when raised solely on G. euwallaceae as well as on Fusarium euwallaceae (Freeman et al., 2015), which is considered to be the primary symbiont. Similarly, we have observed KSHB completing their development on their Fusarium symbiont, and experiments are ongoing to determine if they can feed and reproduce on their Graphium associate (C. Dodge, 2016, unpublished data). The role of Paracremonium pembeum is unknown, and has not been found in association with natural populations of KSHB.

Sibling mating paired with arrhenotokous haplodiploidy, as in the Xyleborini, leads to extremely female-biased sex ratios (Kirkendall, 1993). Female SHB disperse already mated and carrying Fusarium spores in mandibular mycangia to inoculate brood galleries of their own. However, mating pre-dispersal is not a requirement for female SHB since laying an unfertilized egg will produce a son, which she can mate with to produce diploid daughters (Cooperband et al., 2016). Combined, these ecological strategies enable SHB to rapidly colonize new areas (Kirkendall & Jordal, 2006), and the habit of culturing and feeding on fungi rather than directly on plant material allows them to occupy a wide range of hosts (Jordal, 2000).

Although symptoms of Fusarium dieback were recognized much later, the PSHB was first reported in Southern California in 2003 (identified as Euwallacea fornicatus; Rabaglia, Dole & Cognato, 2006). Reports of the KSHB in San Diego County began more recently in 2012 (Eskalen et al., 2012). Since their respective invasions, the PSHB and KSHB together have spread across six counties in California and are also found in adjacent areas of Mexico (García-Avila et al., 2016; for current distributions in California, see http://eskalenlab.ucr.edu/distribution.html). The heart of the PSHB infestation spans Los Angeles and Orange Counties, although they have ranged into neighboring counties as well. KSHB are mostly restricted to San Diego County and Northern Mexico, but several specimens have been collected in other California counties farther north (Santa Barbara and San Luis Obispo Counties).

Over 50 tree species common to Southern California are susceptible to Fusarium dieback, including a variety of urban, riparian, and agricultural hosts (Boland, 2016; Cooperband et al., 2016; Eskalen et al., 2013). The most notable agricultural host is avocado, which has been threatened by the presence of KSHB in San Diego County and PSHB in Ventura County. California produces 90% of domestic avocados, about 70% of which are grown in these two counties (40% in San Diego, 30% in Ventura; California Avocado Commission, 2017). In the 2015–2016 season, avocados comprised a $412 million industry in California (California Avocado Commission, 2017), the third highest crop value in the history of California avocado production. Since the appearance of Fusarium dieback, the avocado industries of Israel and California have faced losses from damage (Freeman et al., 2013b; Mendel et al., 2012) and although the risk seems to be decreasing, the SHB and phytopathogenic Fusarium species continue to pose a threat. The beetle–fungus complex has also caused substantial losses in urban environments, where forced removal of thousands of infested landscape trees has cost millions of dollars over the past few years (University of California, 2015). Additionally, the beetle–fungus complex is invading natural habitats and threatening native plant species. Over a period of six months, disturbance from KSHB resulted in mortality of the majority of native willows in the Tijuana River Valley in San Diego County (Boland, 2016). Willows were the dominant tree species in this riparian habitat that supports numerous plant and animal species, some of which are endangered (Boland, 2016). The spread of SHB and their phytopathogenic fungi therefore have the potential to cause tremendous economic and environmental losses in urban, agricultural, and natural habitats.

Previously, there has been no reliable method of trapping SHB to monitor their distribution and spread. Until recently, the only means of confirming their presence in an area was to find specimens randomly in unbaited Lindgren traps. Here we present the results of 11 field experiments spanning two years, in which we discover and optimize an effective lure for the polyphagous and Kuroshio SHB: the semiochemical quercivorol. We also report other methods of increasing SHB trap catch through trap modifications, as well as the effects of other potential lures. Finally, we test the effects of chemical deterrents on SHB to determine if and to what extent we can repel them in the field.

Methods

Quercivorol

In a field study to screen various semiochemicals for attraction, Synergy Semiochemicals Corp. (Burnaby, BC, Canada) provided a quercivorol lure (Batch #3250) paired with an ultrahigh release (UHR) ethanol bag. Together, they were found to attract the TSHB E. fornicatus in Florida (Carrillo et al., 2015). Due to their close evolutionary relationship to TSHB, we used this lure in an attempt to attract PSHB and KSHB in California. Quercivorol has also recently been used to capture PSHB in Israel (Byers, Maoz & Levi-Zada, 2017).

Quercivorol was first identified from volatiles found in the boring frass of the oak ambrosia beetle Platypus quercivorus (Tokoro et al., 2007), for which it has been identified as an aggregation pheromone (Kashiwagi et al., 2006). It has also been found in odors from artificial diet (made with boxelder sawdust) infested with the associated symbiont of PSHB, Fusarium euwallaceae (M. Cooperband & A. Cossé, 2015, personal communication). Quercivorol ((1S,4R)-p-menth-2-en-1-ol) has two chiral centers (Kashiwagi et al., 2006; Tokoro et al., 2007) and p-menth-2-en-1-ol can have four possible enantiomers. SHB may show varying levels of attraction to these different structural isomers, as has been seen in other scolytines (Byers, 1989; Byers, Maoz & Levi-Zada, 2017).

Experimental design

Experiments were performed in avocado groves in two locations in Southern California: La Habra Heights, Los Angeles County (33°57′33″N, 117°58′10″W) and Escondido, San Diego County (33°08′53″N, 117°01′19″W). Due to their distinct geographical ranges, experiments performed in La Habra Heights targeted PSHB, while experiments performed in Escondido targeted KSHB. Experiments were performed sequentially between the summers of 2014 and 2016.

Black 12-funnel Lindgren traps were used for all experiments and were hung from vertical metal poles 2.5 m in height. Poles were bent to a right angle at the top, and traps were secured to the end of the pole so that they hung freely. To prevent poles from being top heavy, 1 m strips of rebar were hammered into the ground first, and the poles were placed over the rebar to secure them. Traps were spaced roughly 20 m apart and arranged into randomized complete blocks to control for field location. Whenever trap contents were collected, lures were rotated throughout the block to avoid effects of location bias over the course of the experiment. Lures were attached to the second lowest funnel of Lindgren traps. Cups were half-filled with propylene glycol antifreeze to collect, euthanize, and preserve specimens (Allison & Redak, 2017), which were collected weekly or twice weekly for analysis.

Experiment 1: Testing fungal odors

Previous studies have shown certain ambrosia beetles to be attracted to the scent of their fungal symbionts (Hulcr, Mann & Stelinski, 2011; Kuhns et al., 2014). Two novel lures were tested for PSHB attraction: (1) a mixture of their symbiotic fungi F. euwallaceae and G. euwallaceae, grown on an artificial sawdust-based diet medium (modified from Peer & Taborsky (2004)); and (2) a chemical lure consisting of a quercivorol bubble cap (Batch #3250; Synergy Semiochemicals, Burnaby, BC, Canada) and UHR ethanol lure. The diet medium was prepared with sawdust from boxelder, a reproductive host of SHB. About 25 ml of autoclaved medium was poured into a 50 ml plastic Falcon tube and allowed to solidify. Separate, equally concentrated spore suspensions of F. euwallaceae and G. euwallaceae were prepared by the Eskalen lab in the Department of Plant Pathology at the University of California, Riverside, and then combined. About 2 ml of the resulting mixture was used to inoculate diet tubes, which were incubated at room temperature (∼24 °C) for one week before use in Experiment 1. This allowed the fungi enough time to grow over the surface of the diet. The entire fungal-diet mass was removed from each tube in the field and attached to traps using a mesh pocket, to allow fungal scents to escape. Uninoculated diet tubes were prepared and used as a control for SHB attraction to host volatiles in the sawdust. Blank traps served as a negative control. This experiment took place in La Habra Heights for four weeks from August to September 2014 (N = 28, seven replicates of four treatments). Trap contents were collected weekly. Because the exposed artificial diet plugs dry out in the field, fresh inoculated and uninoculated diet tubes were prepared weekly to replace old plugs.

Experiment 2: Effect of ethanol on quercivorol

After noting in a previous experiment that PSHB were not attracted to ethanol lures (C. Dodge & J. Coolidge, 2014, unpublished data), a study was done to determine if the compound quercivorol performs better alone or if paired with the UHR ethanol lure. Experimental traps were baited with a quercivorol bubble cap (Batch #3250), or with a quercivorol bubble cap and UHR ethanol lure. Blank traps served as a control. This experiment was performed in La Habra Heights for six weeks from September to October 2014 (N = 45, 15 replicates of three treatments). Trap contents were collected weekly for analysis.

Experiment 3: Analysis of three different quercivorol blends

Synergy Semiochemicals Corp. provided lures containing two additional ratios of quercivorol and its stereoisomers (trans-p-menthenols) for us to test against the original (Batch #3250). The lure contents differed in ratios of different quercivorol enantiomers. Batch #3250 contained 60% cis/40% trans-p-menthenols (load = 280 mg; release rate = 6 mg/day); Batch #3039 contained 26.7% cis/53.3% trans-p-menthenols, 20% piperitols (load = 290 mg; release rate = 6.5 mg/day); and Batch #3355 contained 11% cis/87% trans-p-menthenols (load = 280 mg; release rate = 7.9 mg/day) (D. Wakarchuk, 2016, personal communication, Synergy Semiochemicals). This experiment was performed in October 2014 in Escondido for three weeks (N = 30, 10 replicates of three treatments). Trap contents were collected weekly.

Experiment 4: Analysis of two additional quercivorol blends

Two additional lures, Batch #3361 and Batch #3362, were provided by Synergy Semiochemicals Corp. for comparison to Batch #3250. Batch #3361 contained 85% cis/15% trans-p-menthenols (load = 280 mg; release rate = 3 mg/day). Batch #3362 contained 57% cis/38% trans-p-menthenols, 5% piperitols (load = 280 mg; release rate = 3 mg/day) (D. Wakarchuk, 2016, personal communication, Synergy Semiochemicals). This experiment was performed in Escondido in March 2015 for two weeks (N = 42, 14 replicates of three treatments). Trap contents were collected twice weekly.

Experiment 5: Batch #3361 vs. P548

ChemTica Internacional (Santo Domingo, Costa Rica) provided quercivorol lures labeled as P548 (68% cis/32% trans-p-menthenols, load = 200 mg; A. Cossé, 2017, personal communication). We tested these against Synergy’s Batch #3361 lure, to see if there was any difference in their attractiveness to SHB. A blank trap served as a control. This experiment took place in Escondido for two weeks from May to June 2015 (N = 30, 10 replicates of three treatments). Trap contents were collected twice weekly.

Experiment 6: Effect of trap cup contents on SHB capture

Because of ease of purchase, we switched to using ethanol-based antifreeze for our experiments. However, due to hot daytime temperatures and dry conditions in the field, evaporation of ethanol-based antifreeze used in the trap cups resulted in poor morphological and molecular insect preservation. A solution containing dimethyl sulfoxide, EDTA, and saturated NaCl, abbreviated DESS, was previously described for high-temperature preservation of DNA in a variety of animals (Yoder et al., 2006). An experiment was performed to see if DESS solution would affect the number of SHB collected from traps, in order to consider its utility as a preservation agent in the field. The trap cup treatments consisted of ethanol-based antifreeze, DESS solution, or an empty (dry) cup. DESS solution was prepared at the University of California, Riverside and transported to the field as a liquid. Trap cups were filled halfway for both the antifreeze and DESS treatments. Two moistened, crumpled Kimwipes were placed in dry cups to dissuade captured insects from flying away. A P548 lure was used for all treatments to attract SHB. This experiment was performed in July 2015 in Escondido for two weeks (N = 30, 10 replicates of three treatments). Trap contents were collected twice weekly.

Experiment 7: Effect of funnel diameter and cup contents on SHB capture

Due to concerns that live beetles could escape the Lindgren trap cups through the hole of the lowest funnel, an experiment was performed to determine if the size of the funnel hole had an effect on the number of SHB collected. In “small” funnel treatments, a plastic funnel with a smaller hole was glued to the rim of the trap cup to reduce the diameter through which trapped SHB could escape. The effect of trap cup collection substrate was also tested. The treatments were as follows: (1) Lindgren funnel traps with no alterations, here called “large” funnel traps, with dry cups; (2) large funnel traps with cups containing DESS solution; and (3) “small” funnel traps with dry cups. A P548 lure was used for all treatments to attract SHB, and crumpled, moistened Kimwipes were placed inside of dry cups. This experiment was performed in Escondido in July 2015 for two weeks (N = 30, 10 replicates of three treatments). Trap contents were collected twice weekly.

Experiment 8: Effect of P548 concentration

The concentration of a lure has been shown in some systems to determine the level of attractiveness to a target insect, ranging from attraction to repulsion (Erbilgin, Powell & Raffa, 2003; Kovanci et al., 2006; Witzgall et al., 2008). We sought to determine whether the concentration of P548 had an effect on level of SHB attraction. In this experiment, one, two, or six identical P548 lures were attached to a trap to determine the attractiveness of different P548 concentrations to SHB. This experiment was performed in Escondido for six weeks between July and September 2015 (N = 30, 10 replicates of three treatments). Trap contents were collected twice weekly.

Experiment 9: Analysis of P548 lures with different release rates

Three P548 lures with varying release rates as described by the company, ChemTica Internacional, were tested. All lures had the same chemical composition and load (200 mg). “P548 A” had the full release rate; “P548 B” had a 50% release rate from that of P548 A; and “P548 C” had a 25% release rate from that of P548 A (C. Oehlschlager, 2016, personal communication, ChemTica Internacional). This experiment took place in Escondido for four weeks between September and October 2015 (N = 30, 10 replicates of three treatments). Trap contents were collected twice weekly.

Experiment 10: Effect of the repellent verbenone

To see if we could repel SHB in the field, ChemTica Internacional provided pouches of Beetleblock Verbenone, a bark and ambrosia beetle repellent. Verbenone has been used in the past to successfully deter economically important bark beetles in the genera Ips and Dendroctonus (Borden, Devlin & Miller, 1991; Fettig et al., 2009), and has more recently been utilized for ambrosia beetle pests (Burbano et al., 2012; Hughes et al., 2017; Jaramillo et al., 2013). We tested the effect of verbenone on SHB by pairing the verbenone pouch with a quercivorol lure (Batch #3361; Synergy Semiochemicals, Burnaby, BC, Canada), to determine if the repellent offset the attractiveness of quercivorol. For a positive control we used a Batch #3361 lure alone, and for a negative control a blank trap was used. This experiment was performed in La Habra Heights for three weeks between October and November 2015 (N = 30, 10 replicates of three treatments). Trap contents were collected weekly.

Experiment 11: Testing verbenone against piperitone

We tested the effects of verbenone against another repellent, piperitone (Synergy Semiochemicals) to determine which deters SHB more effectively. Piperitone was tested because it is the ketone form of the attractant quercivorol, similar to verbenone being the ketone form of the attractant verbenol. This experiment was the first to use piperitone as a repellent against ambrosia beetles. Similar to Experiment 10, both repellents were paired with a quercivorol lure (Batch #3361; Synergy Semiochemicals, Burnaby, BC, Canada) and were tested against a Batch #3361 lure as a positive control. This experiment was performed in La Habra Heights and lasted for six weeks between August and September 2016 (N = 30, 10 replicates of three treatments). Trap contents were collected weekly.

Statistical analysis

Data was collected for each experiment in the form of counts, and were found in all cases to be Poisson overdispersed (Pearson dispersion statistic >1.0). Data were analyzed using a negative binomial regression, using the glm.nb function in the MASS package (Venables & Ripley, 2002) in R to employ a generalized linear model under the assumptions of a negative binomial distribution. The number of SHBs captured was modeled by the effects of treatment, date, and block. Multiple comparisons were performed using Tukey contrasts of least-squares means, using the lsmeans package (Lenth, 2016). To account for outliers, analyses were performed both before and after removing outliers from the data set. Noteworthy effects of outliers are discussed. All analyses were performed using the R free software v3.2.1 (R Core Team, 2015). Results are reported as raw count data. Box plots for each experiment show sample minimum and maximum (horizontal lines at the bottom and top of each plot, respectively) as well as sample median (heavy line inside of box). Upper and lower quartiles are represented by the upper and lower limits of each box, respectively. Data points that fall outside of the quartile ranges are denoted as open circles. Asterisks indicate significance at α = 0.05. Summary statistics and Pearson’s dispersion statistic are also reported for each experiment (Table S1).

Results

Experiment 1: Testing fungal odors

We found that the Batch #3250 quercivorol + UHR ethanol lure attracted significantly more SHB (χ2 = 665.16; df = 3; P < 0.001; Fig. 1A) than either the inoculated or uninoculated diet plug, neither of which were significantly different from our blank control trap (P = 0.876 and 0.729, respectively).

Figure 1 Effect of fungal odors and ethanol on SHB attraction.

Number of shot hole borers collected from traps for each treatment. (A) Experiment 1: The paired Batch #3250 quercivorol + UHR ethanol lure attracted significantly more SHB than either the inoculated or uninoculated diet tubes (P < 0.001), neither of which was significantly different from the blank control trap (P = 0.876 and 0.729, respectively). (B) Experiment 2: The Batch #3250 quercivorol lure by itself attracted significantly more SHB than when paired with a UHR ethanol lure (P < 0.001). Both treatments attracted significantly more SHB than the blank control trap (P < 0.001).

Experiment 2: Effect of ethanol on quercivorol

We found that the Batch #3250 quercivorol lure by itself attracted significantly more SHB than when the lure is paired with a UHR ethanol lure (χ2 = 1221.03; df = 2; P < 0.001; Fig. 1B). Both treatments resulted in significantly higher SHB capture than blank control traps (both P < 0.001).

Experiment 3: Analysis of three different quercivorol blends

We found that the Batch #3250 quercivorol lure, attracted significantly more SHB than Batch #3039 (χ2 = 134.66; df = 2; P < 0.001) and Batch #3355 (P < 0.001; Fig. 2A). Batch #3039 attracted significantly more SHB than Batch #3355 (P < 0.001).

Figure 2 Testing different quercivorol formulations.

Number of shot hole borers collected from traps for each treatment. (A) Experiment 3: Significantly more SHB were attracted to Batch #3250 than to either Batch #3039 (P < 0.001) and Batch #3355 (P < 0.001). Batch #3039 attracted significantly more SHB than Batch #3355 (P < 0.001). (B) Experiment 4: There was no significant difference in number of SHB attracted to Batch #3250 and Batch #3361 (P = 0.427), both of which attracted significantly more SHB than Batch #3362 (P < 0.001).

Experiment 4: Analysis of two additional quercivorol blends

We found no significant difference between Batch #3361 and Batch #3250 (χ2 = 25.97; df = 2; P = 0.427; Fig. 2B). Both of these batches attracted significantly more SHB than Batch #3362 (P < 0.001).

Experiment 5: Batch #3361 vs. P548

We found no significant difference between the number of SHB attracted to the P548 and Batch #3361 quercivorol lures (χ2 = 47.33; df = 2; P = 0.311; Fig. 3). Both of these treatments attracted significantly more SHB than the blank control traps (P < 0.001).

Figure 3 Testing two quercivorols.

Number of shot hole borers collected from traps for each treatment over a two-week period. Experiment 5: There was no significant difference in the number of SHB collected from traps baited with Batch #3361 (Synergy Semiochemicals) or with P548 (ChemTica Internacional; P = 0.311).

Experiment 6: Effect of trap cup contents on SHB capture

We found significantly more SHB in cups containing DESS solution than either in cups with antifreeze (χ2 = 19.53; df = 2; P = 0.010) or dry cups with Kimwipes (P < 0.001; Fig. 4A). There was no significant difference between the number of SHB collected in cups with antifreeze or in dry cups (P = 0.359).

Figure 4 Effect of trap alterations.

Number of shot hole borers collected from traps for each treatment. (A) Experiment 6: Significantly more SHB were collected from trap cups containing DESS solution than either cups with antifreeze (P = 0.010) or dry cups (P < 0.001). (B) Experiment 7: Size of funnel diameter had no effect on number of SHB captured (P = 0.999). Significantly more SHB were collected from cups contained DESS solution than dry cups of either large or small funnel traps (both P = 0.025).

Experiment 7: Effect of funnel diameter and cup contents on SHB capture

When dry cups were used, we found that the diameter size of the funnels had no effect on how many SHB were caught (χ2 = 9.18; df = 2; P = 0.999; Fig. 4B). However, significantly more SHB were collected in cups containing DESS than in dry cups of either large or small funnel traps (both P = 0.025).

Experiment 8: Effect of P548 concentration

Significantly more SHB were attracted to a single P548 lure than to the six-lure treatment (χ2 = 23.14; df = 2; P < 0.001; Fig. 5A). We found no significant difference in the number of SHB captured with the single lure compared to the two-lure treatment (P = 0.259).

Figure 5 Effect of lure concentration and release rate.

Number of shot hole borers collected from traps for each treatment. (A) Experiment 8: Significantly more SHB were attracted to the one- and two-lure treatments than the six-lure treatment (P < 0.001). There was no significant difference between the one- and two-lure treatments (P = 0.259). (B) Experiment 9: Release rate had no significant effect on number of SHB captured (treatment P = 0.315).

Experiment 9: Analysis of P548 lures with different release rates

We found no difference in the number of SHB attracted to P548 A, P548 B, and P548 C with different release rates (χ2 = 3.06; df = 2; treatment effect P = 0.315; Fig. 5B).

Experiment 10: Effect of the repellent verbenone

As expected, the Batch #3361 quercivorol lure as a positive control attracted a significant number of SHB weekly (χ2 = 396.20; df = 2; P < 0.001; Fig. 6A). When paired with a Batch #3361 quercivorol lure, verbenone significantly reduced the number of SHB attracted to the quercivorol lure (P < 0.001), although it still attracted significantly more SHB than the blank control trap (P < 0.001).

Figure 6 Effect of two repellents, verbenone and piperitone.

Number of shot hole borers collected from traps for each treatment. (A) Experiment 10: Verbenone significantly reduced the number of SHB attracted to quercivorol (P < 0.001), although this paired lure still attracted more SHB than the blank control trap (P < 0.001). (B) Experiment 11: When both were paired with quercivorol, significantly fewer SHB were collected from traps with piperitone than traps with verbenone (P < 0.001). Both repellents significantly reduced SHB attraction compared to the quercivorol lure alone (P < 0.001).

Experiment 11: Testing verbenone against piperitone

We found that, when both repellents were paired with a Batch #3361 lure, significantly fewer SHB were collected from traps with piperitone than traps with verbenone (χ2 = 306.47; df = 2; P < 0.001; Fig. 6B). Both repellents significantly reduced the number of SHB attracted to the Batch #3361 quercivorol lure (P < 0.001).

Discussion

Our experiments revealed that lures containing quercivorol were attractive for the capture of PSHB and KSHB. Although quercivorol has been found in odors from sawdust-based artificial diet infested with Fusarium euwallaceae (Cooperband & Cossé, personal communication), we found that SHB were not attracted in the field to diet plugs inoculated with symbiotic fungi in Experiment 1 (Fig. 1A). Fungal volatiles from these lures may have been emitted at concentrations below the threshold of beetle detection under field conditions. We found that removing the UHR ethanol component greatly increased the ability of quercivorol lures to attract SHB (Fig. 1B), suggesting that, unlike many other bark and ambrosia beetles that are attracted to UHR ethanol (Miller & Rabaglia, 2009; Montgomery & Wargo, 1983; Schroeder & Lindelöw, 1989), the PSHB and KSHB had an aversion to UHR ethanol, or that ethanol at that release rate had an antagonistic effect on quercivorol. We have also shown that SHB have an aversion to the repellents verbenone (Fig. 6A) and piperitone (Fig. 6B), which almost completely offset SHB attraction to quercivorol. We found that piperitone is a more effective deterrent for SHB than verbenone (Fig. 6B), making this study the first to demonstrate the potential of piperitone for ambrosia beetle control. Studies with repellents are ongoing to determine optimal release rate, concentration, and effective distance.

Our experiments show that SHB respond differently to different ratios of quercivorol isomers. Both SHB seem to be most attracted to blends where cis quercivorol is the dominant isomer (Fig. 2), and we found no significant difference in their attraction to quercivorol lures from Synergy Semiochemicals Corp. and ChemTica Internacional (Fig. 3). Additionally, we found that SHB do not respond differently to quercivorol lures with different release rates (Fig. 5B). We did find, however, that SHB are more responsive to lower concentrations of quercivorol (Fig. 5A), which attracted significantly more beetles than higher concentration treatments. These findings allow for more cost-effective monitoring, since lures with lower concentrations or release rates are typically less expensive to synthesize and purchase than high concentration, full release rate lures.

Attempts to modify traps to increase SHB catch were somewhat successful. Although alteration of funnels had no effect on the number of SHB being retained (Fig. 4B), we found that using DESS solution as a cup substrate resulted in higher numbers of SHB in trap cups than when ethanol-based antifreeze was used (Fig. 4A). We cannot rule out the possibility that these results were caused by other factors, but this again suggests that SHB have an aversion to ethanol, and also has implications for the use of DESS solution as a field preservation agent.

Because count data is typically skewed, the data were not transformed and are reported as raw count data. However, the possible effect of outliers cannot be ignored. Removing outliers changed significance of the results in one of our experiments. In Experiment 7, the difference between cups with DESS and dry cups was only marginally significant after removing outliers (P = 0.074 and 0.099 for large and small traps, respectively). The effect of date of collection from week to week was also significant in some experiments, which may have influenced the presence of outliers. There are two main explanations for this observation, the first being dosage effects that gradually diminished lure potency over the course of the experiments. This was a known and uncontrollable factor in our experiments, but one that was unlikely to differentially affect our results since all lures had comparable loads and release rates (except in Experiment 9, where the effect of release rate was tested). The second explanation is temperature which, in addition to affecting release rate, would have caused an overall increase or decrease in number of flying SHB (i.e., the pool from which SHB could be collected in the field) and therefore likely would have affected all treatments equally. Thus, the effect of date likely did not affect comparisons between treatments.

Because PSHB and KSHB are members of a closely related species complex with similar fungal symbionts, we assume that their responses to the lures and repellents tested are comparable. Synergy Semiochemical and ChemTica quercivorol lures have both been used in field experiments in La Habra Heights and Escondido, and have resulted in sufficient capture of SHB in each location (C. Dodge, 2016, unpublished data). Both quercivorol lures are also currently used for monitoring purposes across southern California (see http://eskalenlab.ucr.edu/distribution.html), and have resulted in capture of both PSHB and KSHB in their respective locations. It is therefore safe to say that both SHB are attracted to quercivorol; however, there may be differences in their responses to other lures or repellents.

Due to various aspects of their ecologies, bark and ambrosia beetles are notoriously difficult to control. Females spend most of their lives protected within host trees, and disperse already mated with their fungal symbionts. Dispersal typically occurs over a short distance in one of two ways: a flight to another suitable host, or to walk to an unoccupied area of the current host tree. These factors reduce the need for sex or aggregation pheromones in SHB, and indeed none have been discovered. Without the utility of artificially synthesized pheromones or ethanol lures to attract the PSHB and KSHB, the discovery of quercivorol has been a great advance to our knowledge of SHB distribution and spread. Results from our field experiments have greatly optimized SHB trap catch and resulted in an effective monitoring tool for these invasive pests. Monitoring the PSHB and KSHB has previously required field surveys of Fusarium dieback symptoms. Surveys of this kind are time-consuming and rely on accurate and complete visual diagnosis by the surveyor. The development of effective lures provides for passive and less subjective monitoring. Quercivorol could also potentially be used to control SHB through an attract-and-kill type strategy: optimization of both lure and trap could help in decreasing overall SHB population numbers in infested areas, limiting opportunities for the beetle to spread. Paired with effective placement of piperitone or other repellents, this could help to protect uninfested areas from SHB attack.

Supplemental Information

Supplemental Information 1 Summary and dispersion statistics.

Minimum, maximum, and mean values of number of shot hole borers captured per trap for each experiment. Standard error of the mean and Pearson’s dispersion statistic is also reported.

Click here for additional data file.

We would like to thank Synergy Semiochemicals Corp. and ChemTica Internacional for supplying us with lures and keeping us informed about new syntheses to test. We would also like to thank Lupe Hernandez from the Henry Avocado Corporation in Escondido, CA and Raul Alvarado for allowing us to perform these field experiments in their avocado orchards. We gratefully acknowledge Veronica Fernandez, Crystal May Johnston, Amanda Alcaraz, Shannen Hilse, Nickolas Anthony Moreno, Barbara Baker, Augustine de Villa, and Kimberley Garcia for their help with field work and data collection. We would also like to thank Paul Rugman-Jones for input on statistical analyses, the Eskalen lab at UCR for providing fungal isolates, and Stephanie Russell for providing DESS solution.

Additional Information and Declarations

Competing Interests

Author Contributions

Data Availability

The authors declare that they have no competing interests.

Christine Dodge performed the experiments, analyzed the data, contributed reagents/materials/analysis tools, wrote the paper, prepared figures and/or tables, reviewed drafts of the paper.

Jessica Coolidge performed the experiments, reviewed drafts of the paper.

Miriam Cooperband conceived and designed the experiments, reviewed drafts of the paper.

Allard Cossé conceived and designed the experiments, reviewed drafts of the paper.

Daniel Carrillo conceived and designed the experiments, reviewed drafts of the paper.

Richard Stouthamer conceived and designed the experiments, contributed reagents/materials/analysis tools, reviewed drafts of the paper.

The following information was supplied regarding data availability:

Dodge, Christine (2017): R code. figshare.

10.6084/m9.figshare.5021297.v1

Dodge, Christine (2017): Raw data spreadsheets (Excel). figshare.

10.6084/m9.figshare.5021327.v1

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
