# Peer review of "Quercivorol as a lure for the polyphagous and Kuroshio shot hole borers, Euwallacea spp. nr. fornicatus (Coleoptera: Scolytinae), vectors of Fusarium dieback"

_PeerJ, doi:10.7717/peerj.3656_

## Round 0.1 · original submission · Major Revisions

This MS is a well-written and potentially useful addition to the literature in terms of IPM with these exotic pests. The authors describe a number of experiments that should help to set a foundation for further work to refine the various lures and repellents.

I am vacillating between minor and major revisions, but I'd like the chance to send this back to one or more reviewers for a second round if I deem it necessary, so I have opted for major revisions.

In particular the co-authors should pay attention to the following:

-Reviewer 3 brings up some important statistical points. In particular (among the other points) it is not clear how the authors conducted multiple comparisons. This should be made clear. I'd also add that the current method of designating significantly differing means in the figures is rather confusing, and I'd suggest going to a system of letters denoting similar or different means.

-Reviewers 1 and 3 both mention the fact that PSHB and KSHB were targeted in the different experiments conducted in Escondido and La Habra. in reality I believe that most of the work was in Escondido, so the results are most arguably relevant to KSHB. That said, how different are these two insects in terms of their chemical ecology? Do we have any hints? The co-authors give some idea of genetic differences in their introductory paragraphs. As the co-authors know, even within a species semiochemical emissions and responses can differ quite substantially. And this effect can even be seen at times in sympatric same-species assemblages – even on the same host material. The fact that different experiments were conducted on the different subspecies does make things a bit confusing in terms of how universal we might take these results to be. I *suspect* that the authors seeming contention (seeming, because KSHB and PSHB are mainly lumped as SHB in the Discussion) that these results apply universally to both is correct. However a bit more care to making that argument would be useful.

-Several experiments lack a blank (unbaited) control. Due to the nature of the question being asked in each case, and previous experimental results informing the non-blanked experiments, this is not a substantial problem. However acknowledgment of this fact and a rationale in each case in the M&M section would be helpful.

-As noted by Reviewer #2 "It was conspicuous that no information was provided on the composition of the ChemTica lure (P548)." Please rectify this.

-As noted by Reviewer #1 "...there is no discussion of the results from Exp 3 and 4." Please add a discussion of this.

Please see the other helpful comments from the three reviewers and please resubmit a revised MS along with a point-by-point response/rebuttal to the reviewers' comments.

Reviewer 1 ·

Basic reporting

This paper is a good summation of a number of experiments looking at attractants for the polyphagous and Kuroshio shot hole borers. it is structured well with sufficient references, tables, figures and background. paragraph 3 in the Introduction (starting at line 66) is unnecessary, it goes into too much detail on the fungal associates and is not relevant to the article.
The first paragraph in the Methods section may be more appropriate in the Introduction. also the authors should reference a 2017 paper on this subject by Byers et al.

Experimental design

Although the methods section discusses the different ranges of the two target beetles (PSHB and KSHB) and the title indicates attractants for these, the paper and design itself never discuss or even test if there is any difference in response to the different lures. the first paragraph in the methods hints at this but it is never discussed. it would have been very interesting to see if these potentially different species respond differently to the different lures tested. To complete this thought, why weren't some tests conducted in Florida where one of the co-authors is located. experiments 4 and 5 used different "blends' but they were only tested in Escondido on KSHB. Maybe there would have been different responses by PSHB or even TSHB if tested. why wasnt this done? it would have added a lot to the paper. Also with regards to Exp 4 & 5, without a more detailed discussion regarding the enantiomeric composition of these lures (especially P548), i'm not sure it helps the with this paper at all.also, why were these experiments conducted for only 2 weeks? a longer trapping period may have clarified results. for Exp 6 and 7 why were wet Kimwipes used instead of non-pest strips, the usual method in dry cups to prevent beetle escape?

Validity of the findings

The results are interesting and clearly presented in the written section but i had a hard time understanding the figures. . Again, there is no discussion of the results from Exp 3 and 4. why were there differences in the lures? The authors continuously state the response of SHB but why not clarify that the response is KSHB or PSHB?

Additional comments

There is good information in this paper and for the most part well presented but i think it leaves the reader wanting more. Was there any differences in response by PSHB and KSHB? Are the enantiomeric differences important between these SHBs? Are there plans for followup studies on the two repellents? Although they may have shut down attraction in traps, will there be any studies to see if they are effective in infested trees?

·

Basic reporting

In general I found the MS to be well written and easy to read. As a result I have few specific criticisms / suggestions. I did notice in a few instances the order of citations was not consistent (see lines 36-37, 53-54, 250-251). There was one example of an anthropomorphism that detracted from the MS (see line 62, "throttles"). There is some literature re: cup type for forest Coleoptera that could have been referenced. It is self-serving but I recommend the authors see (Allison and Redak 2017. Annu Rev. Entomol. 62: 127-146). Finally, I suggest the authors not use SHB to refer to PSHB and KSHB and rather just refer to them directly to avoid potential confusion of readers.

In general I think the authors developed and executed good experiments and interpreted their results appropriately. I do think that this MS shares a limitation common to this literature, namely because we lack a mechanistic understanding for the results observed, they are of limited relevance beyond the two target species (PSHB and KSHB). From the perspective of development of survey and detection tools for the target species, the MS is very useful. The predictive power for the development of efficient detection and monitoring tools for other / future insects is limited.

Experimental design

The authors clearly identify the objective of developing survey and detection tools for the two species (PSHB and KSHB). In general the Methods and Materials is very clean and easy to follow. It was conspicuous that no information was provided on the composition of the ChemTica lure (P548).

Validity of the findings

No comment.

Additional comments

In general I think this is a well conceived and executed study that has been well written and not over-interpreted.

Reviewer 3 ·

Basic reporting

No comment

Experimental design

I was pleased to see the authors use a negative binomial model which is appropriate for count data rather than shoe-horn the data into a general linear model using data transformation. However, I found several points of the Statistical Analysis methods confusing or contradictory. First, L267-270 states the data were analyzed using a “generalized linear mixed model” with the “glm.nb function in the MASS package.” This particular function does not support both fixed and random effects. Also, which, if any, factors were treated as random effects in the mixed model is not clearly indicated. L270-271 indicates that treatment, date, and block were used as explanatory variables, but all of these appear to have been treated as fixed effects. Thus, I am confused about whether the authors truly performed mixed models as stated. Such models would be appropriate as the time and block variables are natural random factors, but I feel such an approach may not be entirely necessary. Whether mixed models were truly used should be clarified.
Second, the Results for a given experiment indicate each treatment was compared to each other. Yet, the Methods do not mention the use of multiple comparisons to perform such tests and to control for error inflation. Were such tests indeed performed? If so, which test? If not, how was error inflation controlled?
Third, while p-values are reported throughout the Results, the authors did not include the testing statistics. Please include, for example for the negative binomial analyses, the chi-square statistic and its associated degrees of freedom in proper APA format. These should be included to ensure proper interpretation of experiment results. Also, p-values should be reported to only three decimal places. Please make both of these changes throughout the Results.

Validity of the findings

The authors indicate on L143-145 that the PSHB and KSHB have distinct geographical ranges and thus the experiments conducted in the two study locations “targeted” one species or another. The Methods clearly indicate no experiment was conducted in both locations; therefore each experiment tested the response of either PSHB or KSHB. Yet, the language in the Results and Discussion implies that the results of a given experiment extend to both beetles. For example, for Experiment 1, L282-283 reads “lure attracted significantly more SHB …,” with SHB being the collective group of beetles. In the Discussion, L337-338, “the PSHB and KSHB had an aversion to UHR ethanol …,” yet the experiment the authors reference indicates it targeted PSHB and was conducted at a location outside of the KSHB range. Extending experimental results of one beetle to the other does not seem reasonable or appropriate given the experimental designs. Is it reasonable to assume that PSHB and KSHB response similarly to the tested lures and repellents? If so, this justification, along with its supporting literature, should be added to the Methods or Discussion to undue reader confusion. If it is not reasonable to extend the results to both species, the Title and Discussion should be reworded/qualified to indicate this restriction. This issue needs to be addressed as it could undermine the utility of the project findings.

Additional comments

Please see attachment for additional, albeit minor comments.

Annotated reviews are not available for download in order to protect the identity of reviewers who chose to remain anonymous.

---

## Round 0.2 · accepted · Accept

The co-authors have addressed the various reviewer comments well and the MS is now acceptable for publication in PeerJ.

I'd like to request that the release rate for P548 be added to the other information about the lure, in parallel with how other release device specifications were reported. Currently the MS contains "68% cis/32% trans p-menthenols, load = 200mg; Cossé, pers. comm.", but lacks release rate information.

Please consider releasing the review history with this MS as it can he helpful in terms of deeper reading of the paper.

Thank you to the reviewers for their work on this MS, and to the co-authors for submitting a well-organized rebuttal.